# NRF2 activators inhibit influenza A virus replication by interfering with nucleo-cytoplasmic export of viral RNPs in an NRF2-independent manner

Fakhar H. Waqas[1,2], Mahmoud Shehata[3,4☉], Walid A. M. Elgaher[5☉], Antoine Lacour[5,6☉], Naziia Kurmasheva[7], Fabio Begnini[8], Anders E. Kiib[8], Julia Dahlmann[9,10], Chutao Chen[1,2], Andreas Pavlou[6], Thomas B. Poulsen[8], Sylvia Merkert[9,10], Ulrich Martin[9,10], Ruth Olmer[9,10], David Olagnier[7], Anna K. H. Hirsch[5,11], Stephan Pleschka[3,12], Frank Pessler [1,2,13]*

1 Research Group Biomarkers for Infectious Diseases, TWINCORE Centre for Experimental and Clinical Infection Research, Hannover, Germany, 2 Research Group Biomarkers for Infectious Diseases, Helmholtz Centre for Infection Research, Braunschweig, Germany, 3 Institute of Medical Virology, Justus-Liebig-University Giessen, Giessen, Germany, 4 National Research Centre, Giza, Egypt, 5 Helmholtz Institute for Pharmaceutical Research Saarland (HIPS) – Helmholtz Centre for Infection Research (HZI), Saarbrücken, Germany, 6 Institute for Experimental Infection Research, TWINCORE Centre for Experimental and Clinical Infection Research, Hannover, Germany, 7 Department of Biomedicine, Aarhus University, Aarhus, Denmark, 8 Department of Chemistry, Aarhus University, Aarhus, Denmark, 9 Leibniz Research Laboratories for Biotechnology and Artificial Organs (LEBAO), Department of Cardiothoracic, Transplantation and Vascular Surgery (HTTG), REBIRTH-Research Center for Translational and Regenerative Medicine, Hannover Medical School, Hannover, Germany, 10 Biomedical Research in Endstage and Obstructive Lung Disease Hannover (BREATH), German Center for Lung Research (DZL), Hannover, Germany, 11 Department of Pharmacy, Saarland University, Saarbrücken, Germany, 12 German Center for Infection Research, partner site Giessen, Germany, 13 Centre for Individualised Infection Medicine, Hannover, Germany

☉ These authors contributed equally to this work.
* frank.pessler@helmholtz-hzi.de

**Data Availability Statement:** All relevant data are within the manuscript and its Supporting information files.

## Abstract

In addition to antioxidative and anti-inflammatory properties, activators of the cytoprotective nuclear factor erythroid-2-like-2 (NRF2) signaling pathway have antiviral effects, but the underlying antiviral mechanisms are incompletely understood. We evaluated the ability of the NRF2 activators 4-octyl itaconate (4OI), bardoxolone methyl (BARD), sulforaphane (SFN), and the inhibitor of exportin-1 (XPO1)-mediated nuclear export selinexor (SEL) to interfere with influenza virus A/Puerto Rico/8/1934 (H1N1) infection of human cells. All compounds reduced viral titers in supernatants from A549 cells and vascular endothelial cells in the order of efficacy SEL>4OI>BARD = SFN, which correlated with their ability to prevent nucleo-cytoplasmic export of viral nucleoprotein and the host cell protein p53. In contrast, intracellular levels of viral HA mRNA and nucleocapsid protein (NP) were unaffected. Knocking down mRNA encoding KEAP1 (the main inhibitor of NRF2) or inactivating the NFE2L2 gene (which encodes NRF2) revealed that physiologic NRF2 signaling restricts IAV replication. However, the antiviral effect of all compounds was NRF2-independent. Instead, XPO1 knock-down greatly reduced viral titers, and incubation of Calu3 cells with an alkynated 4OI probe demonstrated formation of a covalent complex with XPO1. Ligand–

**Funding:** The study was supported by German Federal Ministry for Science and Education (BMBF) award "COVID-Protect" (01KI20143C; to FP), by the BMBF-funded German Centre for Infection Research (DZIF) partner site Giessen (to SP), by an Alexander-von-Humboldt Foundation Georg Forster Research Fellowship (to MS), and by the European Union's Horizon 2020 research and innovation program under the Marie Skłodowska Curie grant agreement No 860816 (to AKHH). The funders had no role in study design, data collection and analysis, decision to publish, or preparation of the manuscript. FHW received salary support from BMBF award "COVID-Protect" (01KI20143C). MS received salary support from the Alexander-von-Humboldt Foundation Georg Forster Research Fellowship.

**Competing interests:** The authors have declared that no competing interests exist.

target modelling predicted covalent binding of all three NRF2 activators and SEL to the active site of XPO1 involving the critical Cys528. SEL and 4OI manifested the highest binding energies, whereby the 4-octyl tail of 4OI interacted extensively with the hydrophobic groove of XPO1, which binds nuclear export sequences on cargo proteins. Conversely, SEL as well as the three NRF2 activators were predicted to covalently bind the functionally critical Cys151 in KEAP1. Blocking XPO1-mediated nuclear export may, thus, constitute a "noncanonical" mechanism of anti-influenza activity of electrophilic NRF2 activators that can interact with similar cysteine environments at the active sites of XPO1 and KEAP1. Considering the importance of XPO1 function to a variety of pathogenic viruses, compounds that are optimized to inhibit both targets may constitute an important class of broadly active host-directed treatments that embody anti-inflammatory, cytoprotective, and antiviral properties.

## Author summary

Virus infections often cause organ damage via excessive inflammation and oxidative stress. The identification of host-directed treatments that reduce inflammation, accumulation of reactive oxygen species, and viral infectivity is an important goal of antiviral drug development. One advantage of host-directed antivirals is that their targets are encoded by the stable host genome, making emergence of viral resistance less likely. The KEAP1/NRF2 signaling pathway is the most important pathway in humans that protects cells from oxidative stress, and it also induces antiviral and anti-inflammatory responses. NRF2 activators, therefore, are promising candidates for development of host-directed antivirals. We evaluated three NRF2-activating compounds as host-directed treatments for influenza A virus (IAV) infection. All three compounds reduced viral replication, cellular inflammation, and reactive oxygen species. Surprisingly, these effects were completely independent of NRF2 signaling. Instead, we found that these compounds (particularly 4-octyl itaconate) interfere with export of viral RNA/protein complexes from the nucleus, thereby reducing release of viral particles. The most plausible explanation is that the "natural" target of these compounds, KEAP1 (which limits NRF2 signaling), and the nuclear export factor XPO1 (which is required for egress of IAV from the nucleus) contain similar binding sites, thus allowing "NRF2 activators" to also inhibit XPO1.

## Introduction

The current SARS-CoV-2 pandemic has underscored the need for host-directed antiviral compounds. Compared against direct-acting antivirals, inhibitors of host functions that are required for viral replication embody at least two advantages. Firstly, they are likely to have a broader spectrum of activity because more than one viral species may use the same pathway. Secondly, emergence of resistance is less likely because (i) the targets are encoded in the less mutation-prone stable host genome, and (ii) viruses would need to adapt to alternate host pathways or factors in order to become resistant.

Small molecules that activate the cytoprotective and immunomodulatory nuclear factor erythroid 2-related factor 2 (NFE2L2, in short known as NRF2) signaling pathway have received considerable attention because in addition to their well-characterized antioxidative and anti-inflammatory effects they have been shown to interfere with infectivity of both RNA

and DNA viruses [1–10]. A unifying feature of these NRF2 activators is that they possess highly reactive electrophilic groups, such as electron-deficient C = C double bonds that can undergo Michael addition reactions with nucleophilic targets such as sulfhydryl groups on cysteine (Cys) residues. Their designation as NRF2 activators derives from their ability to covalently bind critical Cys residues on the NRF2 inhibitor Kelch-like ECH-associated protein 1 (KEAP1), which allows entry of the transcription factor NRF2 into the nucleus, where it activates the transcription of cytoprotective and antioxidative genes [11]. However, due to their innate electrophilicity, these compounds can potentially interact with nucleophiles on other proteins. On one hand, direct antiviral effects have been postulated. For instance, bardoxolone (BARD) inhibits SARS-CoV-2 replication in Vero and Calu-3 cells, which correlates with its ability to inhibit SARS-CoV-2 3C-like protease by binding to its active site [10]. On the other hand, in most cases a presumably NRF2-dependent modulation of host-cell targets and signaling pathways has been implicated. In the case of sulforaphane (SFN), amplification of antiviral interferon (IFN) responses via NRF2 in infected nasal epithelial cells was suggested as an antiviral mechanism against influenza A virus (IAV) [12] and increased IFN responses due to activation of HMOX1 by NRF2 were part of the mechanisms restricting Dengue virus infection in cell-based models and suckling mice [4]. However, even though the antiviral effect of 4-octyl itaconate (4OI) against SARS-CoV-2 in Vero hTMPRSS2 and Calu-3 cells was partially NRF2-dependent, it did not depend on HMOX1 expression [1]. While the exact mechanisms of antiviral activity of these NRF2-agonists have not been defined, a unifying hypothesis is that they interfere with activity of host factors required for optimal viral replication. However, it remains to be clarified to what extent these host-directed antiviral effects are mediated by induction of NRF2 signaling or whether they result from interactions of the compounds with other targets.

We have recently evaluated itaconic acid and its derivatives dimethyl itaconate and 4OI [2] as well as the itaconate isomers mesaconate and citraconate [9] as host-directed treatments for influenza A virus (IAV) infection. Of note, these compounds (all of which activate NRF2 signaling to varying extent) did not affect levels of viral RNA in host cells but reduced release of infective virions into the supernatant. This was consistent with a block of a post-transcriptional step in IAV replication, for instance at the level of export of viral ribonucleoprotein (vRNP) from nucleus into cytoplasm. Indeed, Sethy et al. had previously found that compounds containing an itaconic acid backbone (which in principle retain the properties of Michael acceptors typical of unmodified itaconate and other NRF2 activators) led to nuclear retention of IAV vRNPs, suggesting inhibition of their nuclear export [13]. Egress of vRNPs out of the nucleus is a required step in the life cycle of IAV. Evidence this far indicates that exportin-1 (XPO1; also known as chromosome region maintenance 1, CRM1) is the most important export factor of IAV vRNP, whereas function of another nuclear export factor, NXF1, is inhibited by IAV nucleoprotein (NP) [14]. In the present work, we tested (i) whether interference with IAV infectivity is a shared feature of 4OI and other well-studied NRF2 activators and (ii) whether an antiviral effect is mediated by inhibiting nuclear export via XPO1. We find that NRF2 activators of greatly differing molecular structure inhibit release of infectious IAV virions from host cells, which is independent of NRF2 signaling but correlates with their ability to delay nuclear–cytoplasmic export of vRNPs. Indeed, we provide (i) biochemical proof of covalent binding of 4OI to XPO1 and (ii) drug–target modeling data suggesting that these NRF2 activators can covalently dock into the active site of XPO1 and that the predicted binding energies correlate with the strength of the observed antiviral effects. Thus, interference with XPO1-mediated nuclear export may be an NRF2-independent mechanism of host-directed antiviral activity of compounds traditionally known as NRF2 activators.

## Results

### NRF2 activators delay nuclear export of influenza A virus vRNPs

We used the respiratory epithelial cell line A549 to test whether 4OI inhibits IAV infectivity by interfering with nuclear export of vRNPs and whether similar effects are seen with the *bona fide* NRF2 activators BARD and SFN and the selective inhibitor of nuclear export SEL (also known as KPT330), a well-characterized direct inhibitor of XPO1 function. Chemical structures of the four compounds are shown in Fig 1, and the experimental layout in Fig 2A.

At 12 h post infection (p.i.) (corresponding to roughly one cycle of viral replication and release), all compounds reduced viral titers in supernatants of infected cells between 8-fold (SFN) and 40-fold (SEL). By 24 h p.i., the reduction by SEL and 4OI was even more pronounced, a lesser reduction (approx. 10-fold) was observed with BARD, but none with SFN (Fig 2B). We then used immunocytochemical staining for viral nucleocapsid protein (NP, the main protein component of IAV vRNPs) to detect infected cells and subcellular localization of viral protein. The percentage of NP+ cells (indicating the fraction of all cells that were infected) was significantly higher under SEL treatment at 4, 6 and 8 h p.i. (S1 Fig), which was partially due to a decline in NP+ cells in the other groups at 6 and 8 h. This effect of SEL may have resulted from enhanced cell entry, protection from cytopathic effects, longer persistence of vRNP in cells, or a combination of the three. In the absence of treatments, about equal proportions of infected cells exhibited either nuclear or cytoplasmic localization of NP by 6 h p.i.,

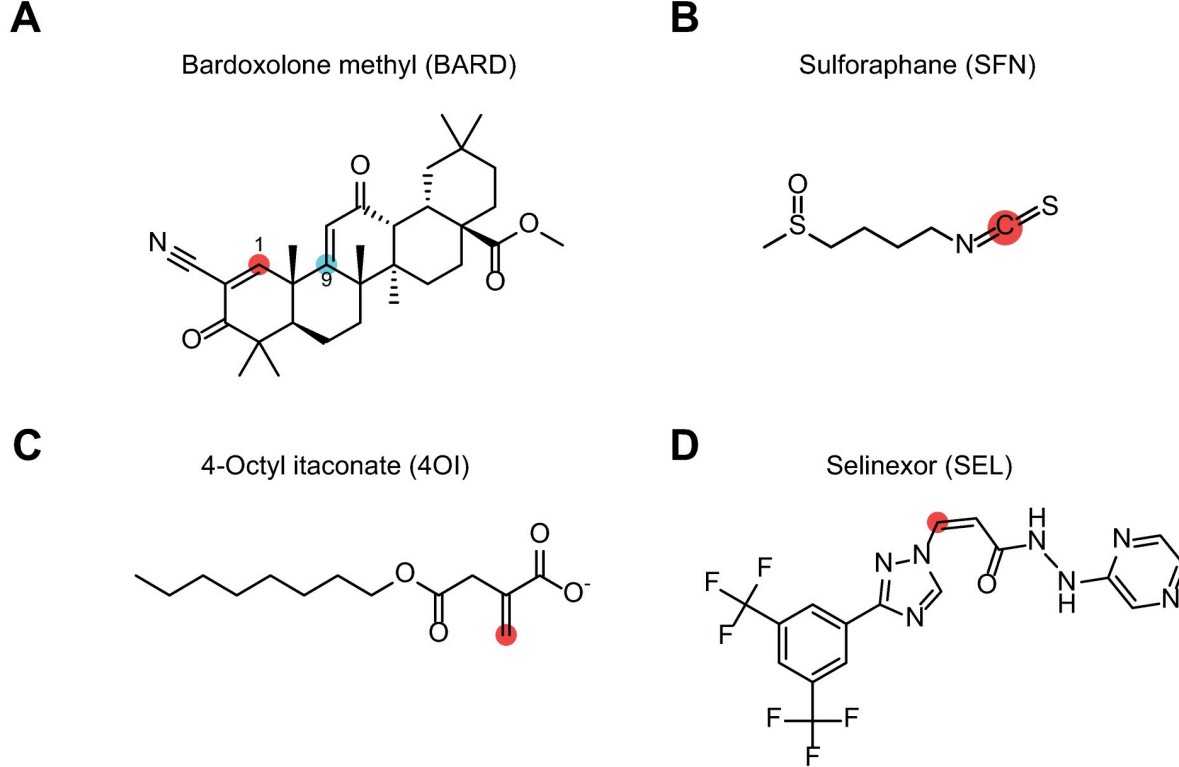

**A** Bardoxolone methyl (BARD)

**B** Sulforaphane (SFN)

**C** 4-Octyl itaconate (4OI)

**D** Selinexor (SEL)

**Fig 1. Chemical structures of the four compounds used.** The reactive electrophilic carbon atoms that can potentially undergo Michael addition from nucleophilic targets are highlighted in red or blue. **A.** Bardoxolone methyl (BARD) is unique in that it has two reactive carbon atoms at positions 1 (red) and 9 (blue). **B.** Sulforaphane (SFN). **C.** 4-Octyl itaconate (4OI). **D.** Selinexor (SEL). This *bona fide* XPO1 inhibitor is not known to be an NRF2 agonist, but also possesses one electrophilic double bond.

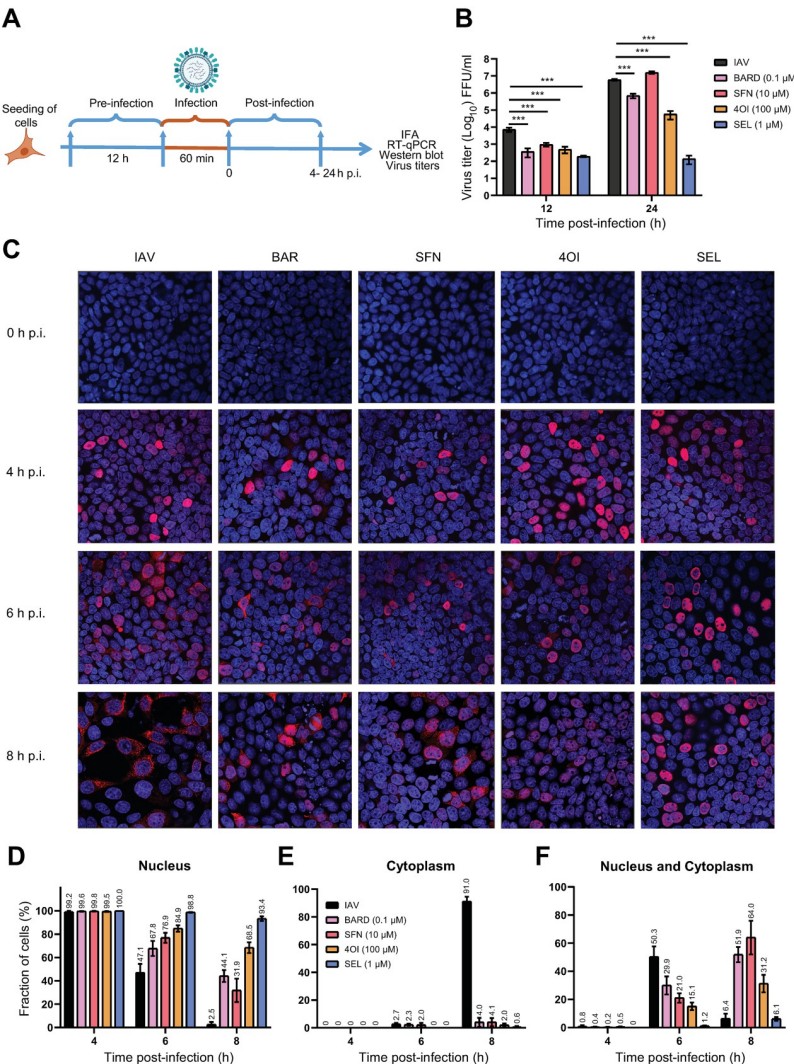

**Fig 2. NRF2 activators reduce release of IAV virions and inhibit nuclear export of vRNP. A. Schematic of experimental layout.** A549 cells were pretreated with the compounds (SEL, 1 µM; 4OI, 100 µM; BARD, 0.1 µM; SFN, 10 µM) for 12 h, were then infected with IAV PR8M (MOI = 0.05 in B, MOI = 1 in C-F) for 1 h and subsequently incubated in fresh buffer containing the compounds. Measurements were performed at the indicated times post infection (p.i.). **B. NRF2 activators reduce release of progeny virions.** Viral titers (FFU/mL) in cell culture supernatants were determined 12 and 24 h p.i. $n$ = 3. **C-F. NRF2 activators interfere with nuclear export of vRNP.** Subcellular localization of viral NP was determined by immunofluorescence 4, 6, and 8 h p.i. Viral NP was visualized by indirect immunofluorescence using Cy3-labeled secondary antibody (561 nm, red) and nuclei by staining DNA with DAPI (405 nm, blue). Cells with NP staining in nucleus, cytoplasm or both nucleus and cytoplasm were quantified by visual inspection. $N$ = 2 replicates, $n$ = 7 digital images per replicate. **C.** Representative microscopic images. **D.** Proportion of cells with nuclear NP staining only. **E.** Proportion of cells with cytoplasmic NP staining only. **F.** Proportion of cells with both nuclear and cytoplasmic NP staining. Data are shown as means ±SEM. One-way ANOVA with Tukey's post-hoc test. $p$ = * ≤0.05, ** ≤0.01, *** ≤0.001.

whereas localization was predominantly cytoplasmic 8 h p.i., thus demonstrating the expected vRNP egress from the nucleus (Fig 2C–2F). All four compounds increased the proportion of cells with exclusively nuclear staining 6 h p.i. and greatly reduced the proportion of cells with exclusively cytoplasmic staining 8 h p.i., whereas a marked increase of cells showing both cytoplasmic and nuclear staining was seen 8 h p.i. under treatment with 4OI, BARD, and SFN.

Overall, SEL exerted the strongest effects, followed (in order of efficacy) by 4OI, BARD, and SFN. Thus, all four compounds interfered with nuclear export of vRNPs and subsequent release of infectious progeny virions, but with varying efficiency.

Considering that the nuclear export protein XPO1 is a well-validated target of SEL, we assessed the contribution of XPO1 to IAV replication and the antiviral effects of the compounds. Knock-down with specific siRNA resulted in a >95% reduction in *XPO1* mRNA and an approximately 85% reduction in XPO1 protein in uninfected and infected A549 cells (Fig 3A–3C). Knock-down of XPO1 did not affect levels of IAV hemagglutinin (HA) mRNA or NP in the absence or presence of the four compounds (Fig 3D and 3E). In contrast, viral titers in supernatants were about 90% lower in XPO1 knock-down than in wild-type cells. The remaining viral activity was consistent with residual XPO1 protein expression seen after knock-down. In the cells transfected with control siRNA, all compounds reduced viral titers, whereby reduction by SEL was by far the greatest (Fig 3F). In the knock-down cells, all compounds led to a further reduction of viral titers, which was consistent with inhibition of residual XPO1 activity (Fig 3F). All compounds reduced intracellular IFN responses, whereby IFN reduction by SEL correlated the least with its antiviral effect and appeared to be greater in the knock-down cells (Fig 3G and 3H). Raised intracellular ROS levels are part of the host response to IAV infection and contribute to cell stress and organ damage, whereas induction of NRF2 signaling is expected to improve redox balance [5]. IAV infection led to a brisk rise in mitochondrial ROS (mROS) levels, which was slightly lower in the XPO1 knock-down cells (Fig 3I). Of note, even though the compounds greatly reduced viral titers, they reduced mROS only modestly, which was independent of XPO1 status. A plausible explanation is that the rise in mROS resulted to a considerable extent from ongoing nonproductive intracellular replication that could proceed despite the block to vRNP export from the nucleus.

IAV infection has been shown to both elevate and depress NRF2 signaling [5,15], and we therefore assessed expression of *NFEL2L2* mRNA (encoding NRF2), *KEAP1* mRNA and four potentially NRF2-regulated mRNAs (Figs 3J and S2). XPO1 knock-down did not affect their expression in uninfected or infected untreated cells. Infection modestly raised *KEAP1* mRNA expression in WT and XPO1 knock-down cells, but tended to downregulate the other analyzed mRNAs except *HMOX1*. The treatments tended to reverse these expression changes independent of XPO1 expression status. HMOX1 had been implicated in antiviral defenses, but its induction did not correlate with the observed antiviral effects in this experiment: its induction was by far the greatest by BARD (which had the weakest antiviral effect) but was only minimal by SEL, which essentially abolished viral release.

## The antiviral effect is NRF2-independent

Considering that the antiviral effect of 4OI, BARD, and SFN is presumably mediated by activation of NRF2 signaling, we tested their impact on IAV infection in cells with a targeted inactivation of the *NFE2L2* gene, which encodes NRF2 protein. We have previously shown that human induced pluripotent stem cells (hiPSC)-derived vascular endothelial cells (ECs) support IAV PR8 (H1N1) infection, but that viral transcription reaches lower levels than in A549 cells [16]. We inactivated the *NFE2L2* gene in hiPSC by CRISPR/Cas9 editing and differentiated wild-type and *NRF2*$^{-/-}$ cells into vascular ECs. As expected, the three NRF2 activators led to a modest induction of *NFE2L2* mRNA levels in wild-type cells, but SEL had the same effect (Fig 4A). *NFE2L2* mRNA was not detected in the *NRF2*$^{-/-}$ cells. Consistent with our previous observations that ECs support a relatively low degree of RNA replication of IAV PR8M [16], viral titers in supernatants of wild-type ECs 24 h p.i. were substantially lower than in A549 cells (7x10$^1$ vs. 2x10$^5$) (Fig 4B). Nonetheless, effects of the interventions were clearly evident.

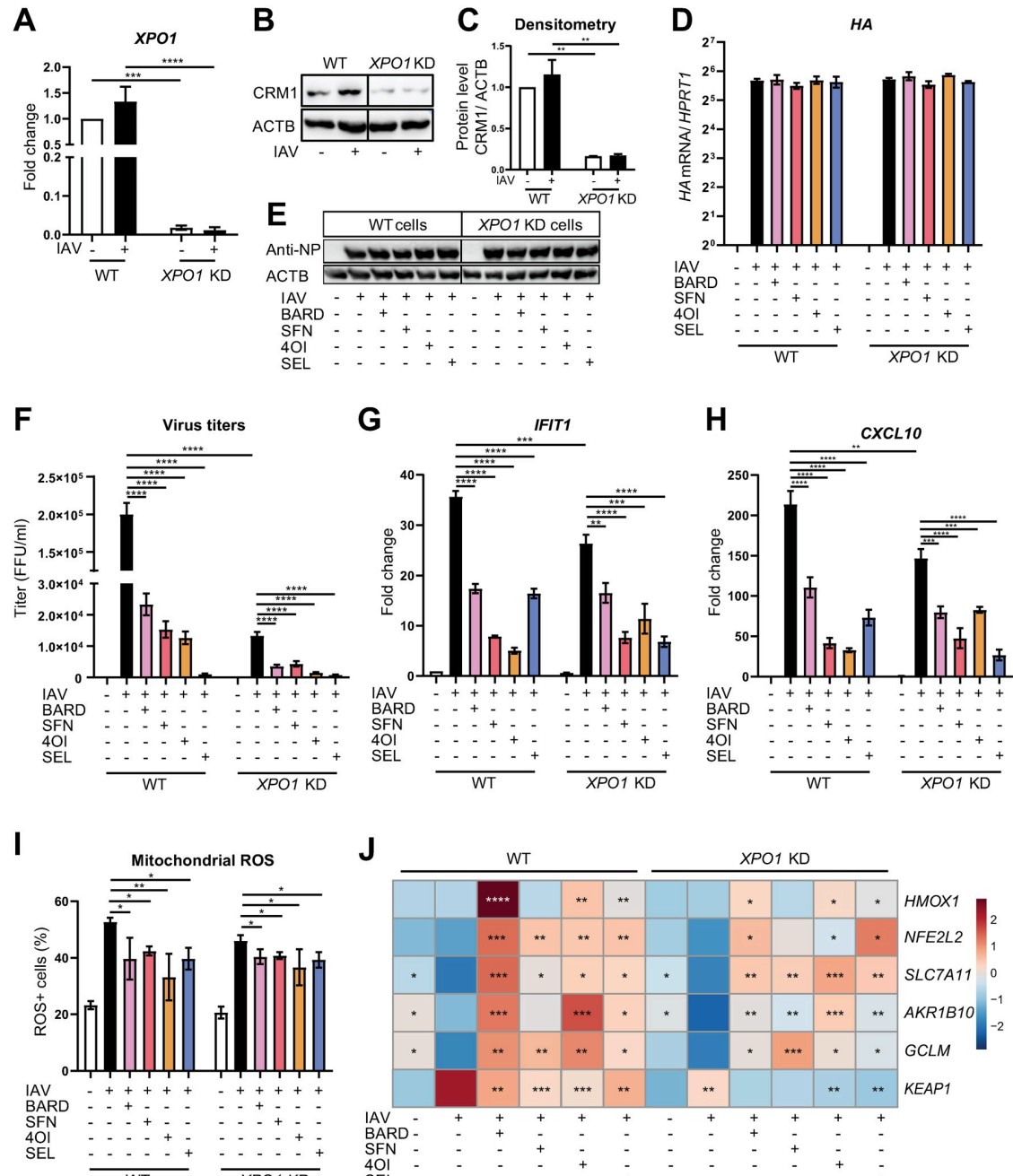

**Fig 3. Effects of XPO1 knock-down on IAV infection, cellular responses, and antiviral activity of the compounds.** A549 cells were transfected for 24 h with specific siRNA targeting XPO1 mRNA or nonspecific siRNA. Cells were then pretreated with the compounds (SEL, 1 μM; 4OI, 100 μM; BARD, 0.1 μM; SFN, 10 μM) for 12 h, infected with IAV PR8M (MOI = 1) for 2 h, and then incubated in fresh buffer containing the compounds for 22 h. **A-C.** Efficiency of XPO1 knock-down. **A.** *XPO1* mRNA (RT-qPCR). **B.** XPO1 protein (immunoblot). **C.** Densitometry of B. **D.** Viral HA mRNA expression with reference to *HPRT1* mRNA as internal control (RT-qPCR). **E.** Viral NP (immunoblot). **F.** IAV titers in cell culture supernatants (foci-forming assay, foci-forming units [FFU]/ml). **G, H.** *IFIT1* and *CXCL10* mRNA (RT-qPCR). **I.** Mitochondrial ROS (flow cytometry). **J.** Expression of *NFE2L2*, *HMOX1*, *SLC7A11*, *AKR1B10*, *GCLM*, and KEAP1 mRNAs (RT-qPCR, internal control *HPRT1* mRNA). The heat map is based on log₂ fold change (scale as indicated in the color legend) with respect to expression in wild-type uninfected cells. Bar graphs for each target gene are shown in S2 Fig for additional clarity. n = 3, means ±SEM. One-way ANOVA with Tukey's post-hoc test, using infected untreated wild-type or knock-down cells as reference. * ≤0.05, ** ≤0.01, *** ≤0.001, **** ≤0.0001.

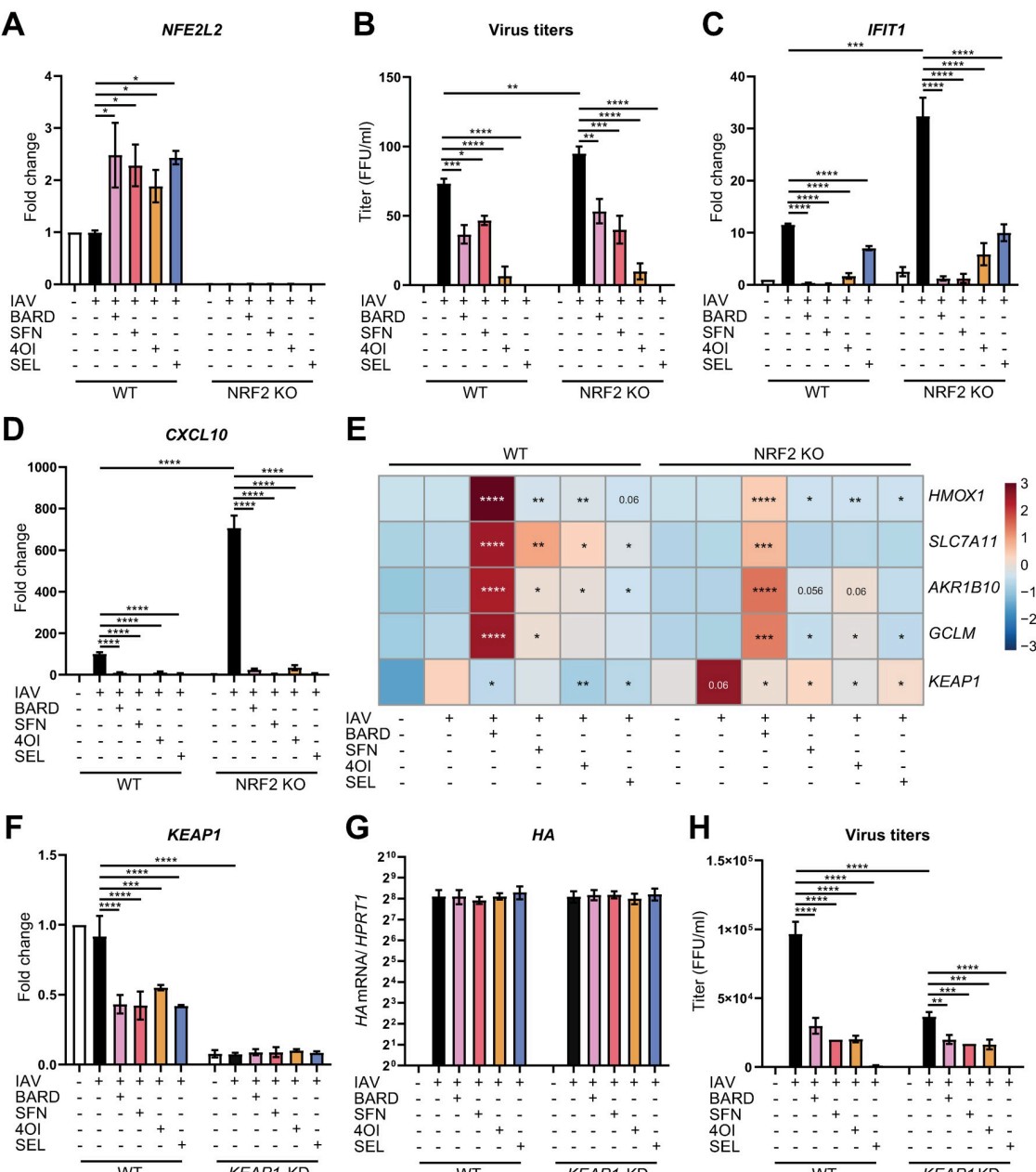

**Fig 4. Antiviral effects of the four compounds are NRF2-independent.** hiPSC-derived wild-type or *NRF2^{-/-}* vascular ECs were pretreated with the compounds (SEL, 1 µM; 4OI, 100 µM; BARD, 0.1 µM; SFN, 10 µM) for 12 h, infected with IAV PR8M (MOI = 1) for 2 h, and then incubated in fresh buffer containing the compounds for 22 h. **A.** *NFE2L2* mRNA (RT-qPCR). **B.** Viral titers in cell culture supernatants (foci-forming assay, FFU/mL). **C, D.** *IFIT1* and *CXCL10* mRNA (RT-qPCR). **E.** Expression of *HMOX1*, *SLC7A11*, *AKR1B10*, *GCLM*, and *KEAP1* mRNAs (RT-qPCR, internal control *HPRT1* mRNA), heat map based on log$_2$ fold change (as indicated in the color legend) with respect to expression in wild-type uninfected cells. Column graphs of these data are shown in S3 Fig for additional clarity. **F-H**, **Knocking down KEAP1 expression reduces viral titers, but does not affect the antiviral effect of the compounds. F**, Expression of KEAP1 mRNA (RT-qPCR). **G**, Expression of viral HA mRNA (RT-qPCR). **H**, Viral titers (foci-forming assay, FFU/ml). *n* = 3, means ±SEM. One-way ANOVA with Tukey's post-hoc test. * ≤0.05, ** ≤0.01, *** ≤0.001, **** ≤0.0001.

Consistent with an antiviral role of NRF2, the *NRF2*$^{-/-}$ ECs supported significantly higher titers than the wild-type cells, which was paralleled by brisker interferon responses (Fig 4B–4D). All compounds reduced viral titers in the same order of efficacy as in A549 cells, i.e. SEL > 4OI > BARD = SFN. Strikingly, pronounced titer reductions of nearly the same magnitude were seen with the *NRF2*$^{-/-}$ cells, demonstrating that the compounds' antiviral effect did not require NRF2 signaling and was likely mediated by interaction(s) with a different common target(s). Baseline expression of *KEAP1* mRNA was higher in *NRF2*$^{-/-}$ cells (Figs 4E and S3). Infection modestly increased *KEAP1* mRNA in both WT and *NRF2*$^{-/-}$ cells, and treatment with all compounds led to a significant reduction, which appeared to be more pronounced in the *NRF2*$^{-/-}$ cells. All three NRF2 activators led to a stronger induction of potentially NRF2-related genes, particularly *HMOX1*, in ECs than in A549 cells, which might be due to lower baseline expression of these genes in ECs compared to A549 cell (S4 Fig). Loss of NRF2 had differential effects on the four potentially NRF2-regulated antioxidative genes in that induction of *HMOX1* and *SLC7A11* by SFN, 4OI and SEL was abrogated, induction of all 4 target genes by BARD was attenuated, whereas NRF2 knock-out did not have a notable effect on induction of *AKR1B10* and *GCLM* mRNA by SFN, 4OI and SEL (Figs 4E and S3). Taken together, these results suggest that the antiviral effects of the four compounds are largely independent of the presence of intact NRF2-signaling and do not correlate with their ability to induce expression of potential NRF2 target genes including *HMOX1*. In addition, these data demonstrated that NRF2 signaling restricts IAV replication in untreated ECs. We therefore tested whether augmenting NRF2 signaling by knocking-down expression of its inhibitor KEAP1 in A549 cells would potentiate the anti-influenza effect. Transfecting cells with specific siRNA against KEAP1 mRNA led to a >90% reduction of KEAP1 mRNA and protein (Figs 4F and S5). Whereas there was no difference in viral HA mRNA levels, viral titers were significantly lower in supernatants from KEAP1 knock down cells than from cells transfected with nonspecific siRNA, suggesting that KEAP1/NRF2 signaling interferes with a post-transcriptional step in the viral life cycle (Fig 4G and 4H). The antiviral effects of all compounds were independent of KEAP1 expression status, demonstrating that the absence of KEAP1 as a competing target did not augment interference with XPO1 function. To test whether XPO1 and NRF2 interact functionally in physiologic or pharmacologically-induced anti-IAV responses, we knocked down *XPO1* mRNA in *NRF2*$^{-/-}$ ECs. Transfection with specific siRNA resulted in a 70% reduction of *XPO1* mRNA independent of *NRF2* genotype (S6 Fig). This experiment confirmed that viral replication was higher in *NRF2*$^{-/-}$ ECs and that the antiviral effects of all compounds were NRF2-independent (S6C Fig). In the double *NRF2* knock-out / *XPO1* knock-down, viral replication was lower than in wild-type cells, suggesting that the proviral effect of XPO1 is stronger than the antiviral effect of NRF2. Curiously, treatment with BARD or SFN did not reduce viral titers further, whereas 4OI and SEL treatment nearly abolished viral replication. These results suggested either that BARD and SFN require NRF2 function in order to inhibit residual XPO1 function in the knock-down cells or that 4OI and SEL target an additional factor whose importance to IAV replication becomes apparent when XPO1 function is reduced in cells lacking NRF2. All compounds reduced *IFIT1* expression independent of NRF2 or XPO1 status. However, there were no strong correlations with the compounds' ability to reduce viral titers (S6D Fig), probably because the observed downregulation reflected a combination of antiviral and anti-IFN mechanisms.

## Binding of the NRF2 activators to the NES-binding site of XPO1

We then tested whether, like *bona fide* inhibitors of nuclear export, 4OI can covalently interact with the active site of XPO1. Indeed, in Calu-3 cells an alkynated bio-orthogonal probe based

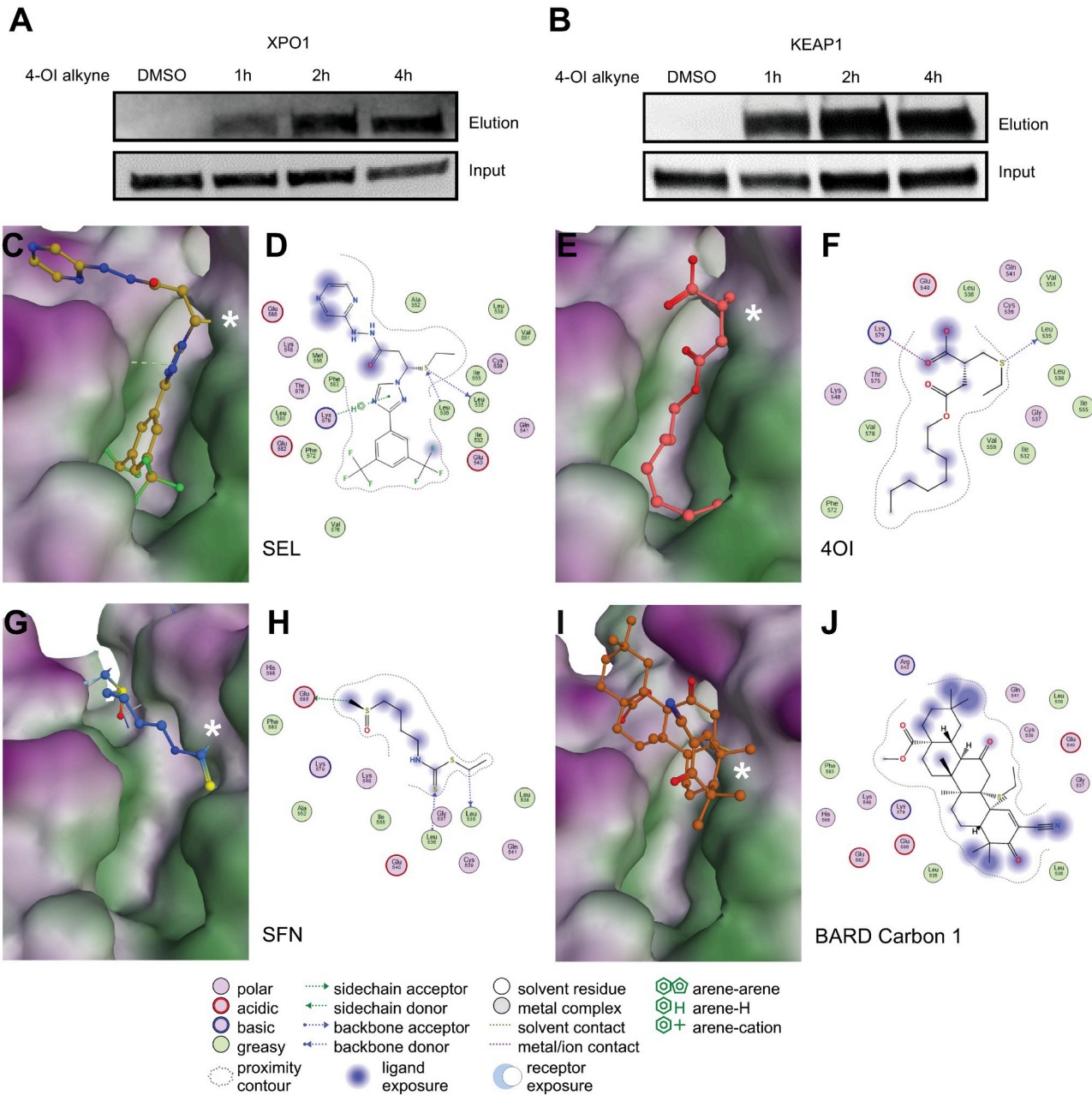

**Fig 5. Biochemical and predicted ligand-target interactions with XPO1. A,B.** "Click-chemistry" pull-down assay demonstrating covalent binding of an alkynated 4OI probe (4-OI-alk) to XPO1 (A) and KEAP1 (B) in Calu-3 cells. At the indicated time points after addition of the probe to the cells, proteins complexed with the probe were detected by immunoblot for XPO1 or KEAP1. **C-J.** Ligand-target modeling studies of the compounds with the active site of XPO1 containing the functionally critical Cys528 (marked with a white asterisk *). Predicted binding energies are shown in Table 1. 3D models and the corresponding 2D interaction diagrams are shown in A,B (SEL), C,D (4OI), E,F (SFN), and G,H (BARD C1). A more detailed binding pose of 4OI to this site, as well as superimposed binding poses of 4OI and leptomycin B, are shown in S9 Fig. * = Cys528.

on 4OI [17] formed complexes with XPO1 and the NRF2 inhibitor KEAP1 (which was used as positive control) within 2 h of incubating the cells in medium containing the probe (Fig 5A and 5B). Preincubating the cells with SEL (unlabeled) efficiently attenuated formation of this complex, indicating that 4OI and SEL target the same site on XPO1 (S7 and S8 Figs). On cargo

**Table 1. Covalent binding energies of the compounds to human XPO1 (CRM1).**

| | Crystal structures used | | |
|---|---|---|---|
| Protein database (PDB) ID | 7L5E | 4HAT | 6TVO |
| Ligand in crystal structure | Selinexor | Leptomycin B | Leptomycin B |
| **Compound** | **Binding energy score (S) (kcal/mol) [a]** | | |
| Selinexor | −6.80 | −6.49 | −5.78 |
| 4-Octyl itaconate | −5.05 | −4.98 | −4.8 to −5.6 |
| Sulforaphane | −4.10 | −4.22 | −4.09 |
| Bardoxolone methyl (reaction at C9) | −1.88 | −1.12 | −3.70 |
| Bardoxolone methyl (reaction at C1) | _ [b] | _ [b] | _ [b] |

[a] Compounds arranged according to descending score values

[b] Docking unsuccessful.

proteins, XPO1 recognizes a nuclear export signal (NES), which is a hydrophobic consensus sequence that binds to a hydrophobic groove of XPO1 located between the HEAT repeats (i.e., structural motifs comprising two alpha helices joined by a short loop) H11 and H12. Most structurally characterized CRM1 inhibitors such as leptomycin B and SEL interfere with NES binding by occupancy of the NES-binding groove via covalent conjugation to the functionally critical Cys528 [18,19]. To obtain additional evidence that 4OI (and possibly BARD and SFN as well) targets this NES binding site, we used ligand-target modelling to obtain a detailed view of predicted interactions between the four compounds and this site. We compared the predicted binding energies of the compounds to the XPO1 active site based on three ligand-XPO1 crystal structures: one co-crystalized with SEL (Protein Database [PDB] ID 7L5E [20]) and two with leptomycin B (4HAT [18] and 6TVO [21]). SEL yielded the best predicted binding energy across all three structures (Table 1) and its predicted binding pose in the 7L5E structure closely matched the crystallographic binding mode (Fig 5C and 5D). Remarkably, 4OI was predicted to undergo extensive interactions with this site: a covalent Michael 1,4-addition reaction with Cys528, H-bonds with Lys537 and Lys568, and extensive interactions of its 8-carbon "tail" within the hydrophobic groove (Fig 5E and 5F, also see the superimposition with leptomycin B shown in S9 Fig). SFN, and, less so, BARD were predicted to bind into this active site as well (**5G-J**). However, binding energies were weaker, which was most likely due to their less extensive interactions with the hydrophobic groove. BARD has two potentially reactive carbons, C1 and C9, but a binding pose could be generated only when forcing the reaction with C9. The NES binding site is relatively narrow. In spite of its hydrophobic nature, BARD could not form higher energy interactions with the hydrophobic groove due to steric hindrance. The predicted binding energies roughly correlated with the ability of the compounds to inhibit nuclear export of viral NP/vRNPs and release of infectious virions into the medium (see also Figs 2 and 4B). This was clearest in the case of SEL, which had the highest binding energy and inhibited nuclear vRNP export and reduced viral titers to, by far, the greatest extent. Taken together, these results strongly suggest that nuclear retention of NP/vRNPs, and subsequent reduction of release of progeny virions, resulted from the compounds blocking the NES binding site of XPO1.

## Binding of SEL and the NRF2 activators to the BTB domain of KEAP1

Having found that the three NRF2 activators are predicted to bind the XPO1 active site, we then tested whether the XPO1 inhibitor SEL could, in turn, interact with KEAP1. In the absence of electrophilic cell stress, KEAP1 binds NRF2 in the cytoplasm and targets it for

**Table 2. Covalent binding energies of the compounds to the BTB domain of human KEAP1.**

|  | Crystal structure used |
| --- | --- |
| Protein database (PDB) ID | 4CXT |
| Ligand in crystal structure | CDDO [a] (Bardoxolone) |
| **Compound** | **Binding energy score (S) (kcal/mol) [b]** |
| 4-Octyl itaconate | −5.01 |
| Selinexor | −4.87 |
| Bardoxolone methyl (reaction at C1) | −4.42 |
| Bardoxolone methyl (reaction at C9) | −3.83 |
| Sulforaphane | −3.68 |

[a] 2-Cyano-3,12-dioxo-oleana-1,9(11)-dien-28-oic acid

[b] Compounds arranged according to descending score values

destruction. Michael addition reactions of electrophilic compounds (including BARD, SFN and 4OI) with Cys residues on KEAP1 interfere with its binding to NRF2, thus allowing entry of this transcription factor into the nucleus [11]. It was not feasible to model all potential interactions of the compounds with KEAP1 because it contains several Cys residues. We focused on the binding site of unmethylated BARD in the BTB domain, which contains the functionally important Cys151 and is available as a co-crystal structure with this ligand [22]. As expected, the three NRF2 activators were predicted to bind this site, whereby 4OI had the strongest binding energy (Table 2). Of note, SEL was predicted to bind this site with an energy intermediate between 4OI and BARD. Like the other compounds, SEL was predicted to undergo Michael addition with Cys151, with additional noncovalent interactions further contributing to binding (Fig 6). Nonetheless, predicted SEL binding to KEAP1 was not as strong as to XPO1 (Tables 1 and 2). In contrast to XPO1, binding poses of BARD could be obtained with both C1 and C9, with the pose based on reaction at C1 showing slightly better predicted binding energy than the C9 pose (Fig 6G–6J). We therefore suggest that BARD is a better ligand for KEAP1 than for XPO1, whereas SEL is a better ligand for XPO1 than for KEAP1. Indeed, the lower affinity of SEL for KEAP1 than for XPO1 is supported by the competition experiments in S7 Fig, where SEL competed more efficiently against formation of the 4OI-XPO1 complex than the 4OI-KEAP1 complex. Taken together, these results suggest that the three studied NRF2 activators and the XPO1 antagonist SEL can potentially interact with the active site(s) in either target, whereby binding is primarily determined by their ability to undergo Michael addition with susceptible Cys residues in the active site and is strengthened by noncovalent interactions.

## The compounds increase nuclear retention of p53 in IAV infected cells

To test whether these compounds also affect subcellular localization of a cellular protein whose function is regulated by XPO1-dependent nuclear-cytoplasmic shuttling [23], we determined nuclear vs. cytoplasmic localization of p53 in the presence or absence of the compounds (S10 Fig). Compared to uninfected cells, IAV infection increased both nuclear and cytoplasmic p53 expression (S10A Fig). Treatment with SEL and, to a somewhat lesser extent, 4OI resulted in a further increase in staining intensity, and nearly all cells now exhibited nuclear p53 staining. In contrast, the impact of BARD and SFN was much less pronounced, i.e. 34.3% (BARD) and 33.8% (SFN) compared to 18.1% cells with nuclear p53 staining in untreated infected cells (S10B Fig).

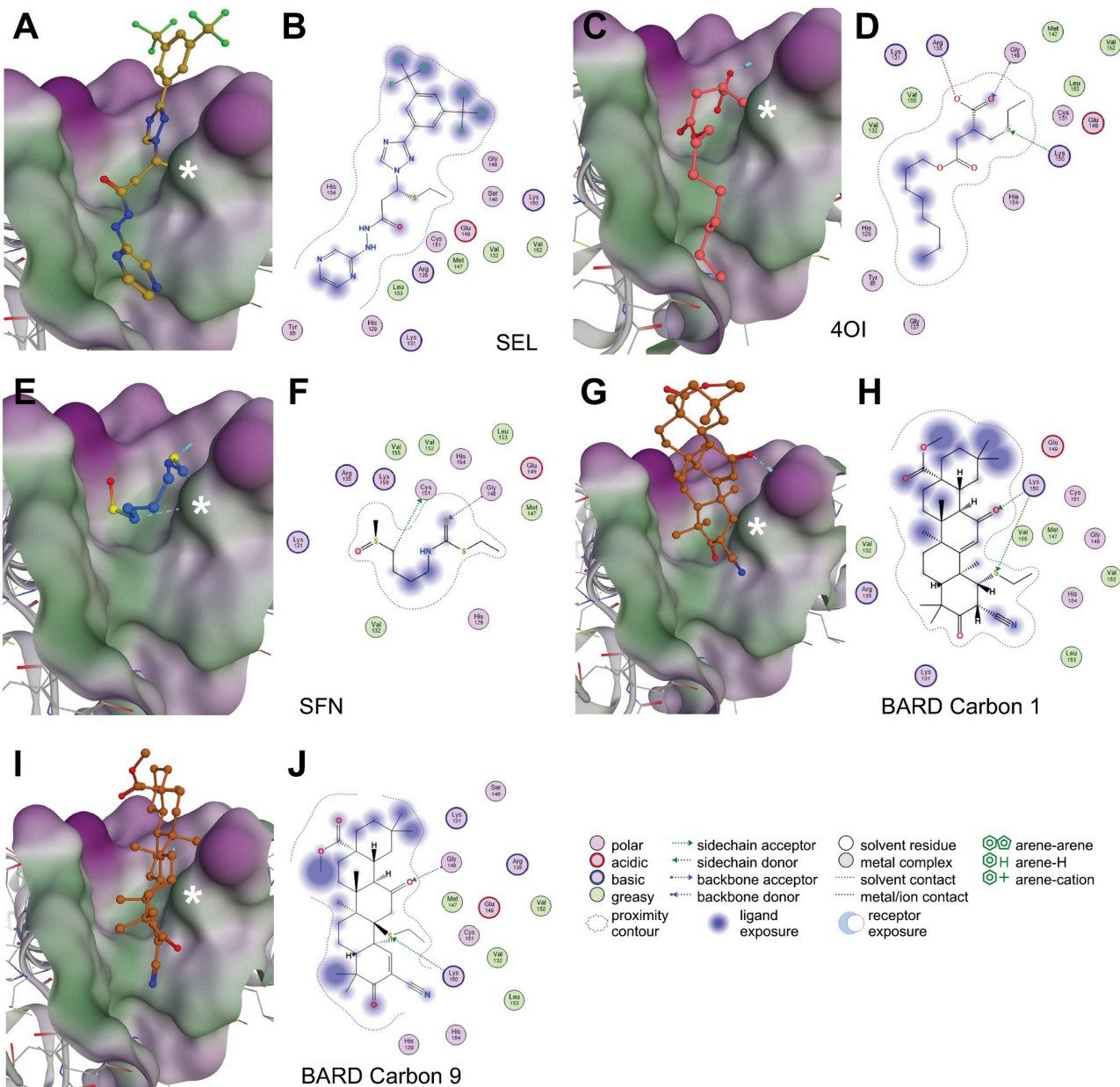

**Fig 6. Predicted ligand-target interactions of the four compounds with the BTB domain of KEAP1.** Ligand-target modeling studies of the compounds with the active site of the BTB domain of KEAP1 containing the functionally critical Cys151 (marked with a white asterisk *). Predicted binding energies are shown in Table 2. 3D models and the corresponding 2D interaction diagrams are shown in **A,B** (SEL); **C,D** (4OI); **E,F** (SFN); **G,H** (BARD C1); and **I,J** (BARD C9). * = Cys151.

## Discussion

### XPO1 as previously unrecognized target of NRF2 activators

In this study based on IAV infection of human cells, we provide comprehensive evidence that inhibition of XPO1-mediated nuclear export of vRNPs is a major NRF2-independent antiviral mechanism of compounds which are traditionally known as NRF2 activators. NRF2 activators induce NRF2 signaling by covalently binding to nucleophilic sulfhydryl groups on the NRF2

inhibitor KEAP1. As shown in Fig 1, the three tested NRF2 activators all possess electrophilic double bonds. From a chemical point of view, it is not surprising that they can react with nucleophiles on other polypeptides if presented in the proper context. Our study, thus, presents evidence that this certain degree of "electrophilic promiscuity" can lead to a biologically highly relevant effect that is mediated by a target other than the originally intended one. Xenobiotic substances with antiviral and/or anti-inflammatory effects are often believed to act by inducing NRF2 signaling, and such substances are also active against viruses that do not have a replication phase in the nucleus [24]. In the case of the NRF2 activator epigallocatechin gallate, it was shown that its anti-IAV activity only partially depended on NRF2 function [12], raising the possibility that it also recognizes other antiviral targets. Future studies should, therefore, explore to what extent activators of NRF2 signaling inhibit influenza or other viruses via XPO1, KEAP1, a combination of the two, or yet other targets.

## Blocking XPO1 as major mechanism of action of 4OI's anti-IAV effect

The various biological effects of 4OI were initially ascribed to inhibition of KEAP1, but subsequent studies have shown that 4OI can also interact with other polypeptides (reviewed in [25]). For instance, it has been suggested that it inhibits aerobic glycolysis by covalently binding to Cys22 of glyceraldehyde 3-phosphate dehydrogenase (GAPDH) [26]. Inhibition of succinate dehydrogenase (SDH) has been suggested to mediate the antiviral effect of 4OI [3], but more recent studies have shown that 4OI does not inhibit this enzyme [27], which agrees with our findings that unmodified itaconate binds to the active site of SDH by noncovalent interactions and not by Michael addition [9]. 4OI is not expected to bind this site efficiently, as the 8-carbon tail reduces potential ionic interactions and provides steric hindrance. In our previous work on itaconates and IAV infection, we noted that itaconates reduced release of infectious influenza A virions but did not affect viral RNA levels [2]. The current data now strongly suggest that 4OI inhibits replication of IAV by interfering with nuclear export of vRNPs, which is a particularly critical step in the life cycle of this virus. The 8-carbon tail of 4OI was originally believed to amplify biological effects of itaconate by facilitating cell entry and increasing electrophilicity [28]. Our ligand-target models strongly suggest that an additional major consequence of the added carbon tail is to optimize target selectivity by enhancing binding to polypeptides that harbor hydrophobic motifs, as exemplified by the hydrophobic groove of XPO1. Current concepts of 4OI binding to cellular targets are based on biochemical assays, e.g., using alkyne-tagged probes [17]. It will now be important to apply the tools of structural biology to test to what extent binding to 4OI targets is optimized by interactions with its 4-octyl tail. This hydrophobic chain was added to native itaconate at the expense of the C4 carboxyl group, thereby reducing the spectrum of interactions with target proteins via ionic and hydrogen bonding. As shown in S10 Fig, 4OI strongly increased nuclear expression of p53. This multifunctional protein can have both protective and pro-apoptotic effects in IAV infection [29]. The functional consequences of p53 modulation by 4OI for influenza virus infection therefore require further study. BARD and SFN inhibited IAV replication to a much lesser extent than 4OI, which agrees well with their less optimal predicted binding to XPO1. It is, therefore, to be expected that they would prove less effective than 4OI to inhibit viruses that depend on XPO1 for optimal replication.

## KEAP1 as possible alternative target of SEL

XPO1 inhibitors have been shown to augment NRF2 signaling by increasing nuclear retention of NRF2 [30–32]. It was hypothesized that this was due to inhibition of nuclear export of NRF2 protein. Our drug-target model suggests that SEL can bind the critical Cys151 in the

BTB domain of KEAP1, but it had weaker effects on expression of antioxidative, potentially NRF2-regulated genes than the other compounds studied herein. We did not model binding of other XPO1 inhibitors to KEAP1, and it is not clear whether binding of XPO1 inhibitors to KEAP1 is pharmacologically relevant. Further studies are, therefore, required to test the hypothesis that activation of NRF2 signaling by binding to (and thus inhibiting) KEAP1 is a noncanonical function of some XPO1 inhibitors.

## XPO1 as antiviral target of selective inhibitors of nuclear export

Of the four tested compounds, SEL reduced nuclear export of vRNPs and release of progeny virions the most, thus underscoring the importance of XPO1-mediated nuclear export for replication of IAV. Building on the observation that the natural XPO1 inhibitor leptomycin B inhibited nuclear vRNP export and replication of IAV *in vitro* [33], it was shown that the synthetic XPO1 inhibitor verdinexor inhibited replication of IAV *in vitro* and reduced viral burden, inflammation, and lung pathology *in vivo* [34,35]. XPO1 function is important for optimal replication of a broad spectrum of human pathogenic viruses, even RNA viruses that do not complete part of their life cycle in the nucleus (S1 Table). As shown by the remarkable increase in nuclear expression of p53 (S10 Fig), XPO1 inhibitors such as SEL conceivably affect infectivity of such viruses by favoring nuclear retention of host cell factors. Indeed, SEL has been reported to inhibit SARS-CoV-2 infection *in vitro* and *in vivo*, whereby nuclear retention of ACE2, the major receptor for SARS-CoV-2, was postulated as a potential mechanism [36]. SEL and verdinexor were originally developed for the treatment of cancer and are licensed for treatment of advanced multiple myeloma and canine lymphoma, respectively. Our results strongly suggest that this class of drugs, termed selective inhibitors of nuclear export (SINE), merit further clinical development as treatments for influenza viruses, coronaviruses and other viral pathogens that have evolved to usurp XPO1 function for part of their life cycle.

## Anti-influenza effects of physiologic KEAP1/NRF2 signaling

Yageta et al. first reported a protective effect of physiologic NRF2 function in a mouse model of IAV infection [37]. Using primary human nasal epithelial cells, Kesic et al. subsequently found that knocking down NRF2 expression with shRNA increased both entry and replication of influenza A/Bangkog/1/79 [12]. Overexpressing NRF2 reduced replication of IAV PR8 in alveolar epithelial cells [38]. These authors also measured HMOX1 expression. Of note, HMOX1 expression rose greatly during IAV infection, but actually decreased when NRF2 was overexpressed and viral replication was restricted. In our study, the proviral phenotype of the *NRF2*$^{-/-}$ ECs and antiviral phenotype of the KEAP1 knock-down cells underscore the importance of NRF2 signaling to cellular defenses against IAV infection, and we provide evidence that NRF2 signaling interferes with a post-transcriptional step in IAV replication. HMOX1 has been considered a key mediator of antiviral mechanisms in the context of NRF2 signaling and oxidative stress responses in general. However, like Kosmider et al. [38], we did not observe a correlation between viral suppression and HMOX1 expression: BARD tended to have the weakest antiviral effect but upregulated *HMOX1* mRNA the most, which was independent of host cell *NRF2* genotype. Thus, the mechanism(s) of viral restriction by NRF2 signaling remain(s) to be defined.

## Methods and materials

### Compounds

Bardoxolone methyl (BARD; 2-cyano-3,12-dioxo-oleana-1,9(11)-dien-28-oic acid methyl ester) was obtained from Hölzel/MedChem Express, Sollentuna, Sweden, (HY-13324), 4-octyl

itaconate (4OI) from Biomol (Cayman chemical) Hamburg, Germany (Cay25374-25), sulfo-raphane (SFN; (*R*)-1-Isothiocyanato-4-(methylsulfinyl)-butane) from Santa Cruz, Heidelberg, Germany (sc-203099), and selinexor (SEL; selective inhibitor of nuclear export KPT-330, from Tebu-Bio, Offenbach, Germany (10-4011-25mg). The compounds were applied at commonly used concentrations that had previously been shown to be nontoxic; 4OI 100 μM [2]; SEL 1 μM [39]; BARD 0.1 μM [40], and SFN 10 μM [41].

## Viruses

Influenza A virus (A/Puerto Rico/8/1934 [H1N1]), here referred to as PR8M for brevity sake, was propagated by plasmid rescue in MDCK-II cells for 48 h at 37˚C. Viral titers in superna-tants were determined by focus-forming unit (FFU) assay, and the viral stocks were then stored in aliquots at -80˚C until use.

## Cells and cell culture

A549 human adenomacarcinoma cells were originally obtained from German Collection of Microorganisms and Cell Cultures GmbH (DSMZ), Braunschweig, Germany and were propa-gated in DMEM medium supplemented with 10% FCS and 2 mM L-glutamine. To generate *NRF2*$^{-/-}$ iPSC, the *NFE2L2* gene was inactivated by CRISPR/Cas9 editing in iPSC line MHHi001-A [42]. Absence of *NFE2L2* mRNA was verified by RT-qPCR (Fig 3A) and absence of NRF2 protein by immunoblot [42]. Wild-type and *NRF2*$^{-/-}$ iPSC were differentiated into vascular ECs using an established protocol, with positive selection of CD31+ cells as final step [43]. hiPSC-ECs were cultured on fibronectin (Corning, New York, USA) coated plates in ECGM-2 medium (PromoCell, Darmstadt, Germany). For XPO1 knock-down, cells were grown to 90% confluency and transfected with specific (ON-TARGETplus Human XPO1 (7514) siRNA—SMARTpool, 5 nmol, L-003030-00-0005, Horizon Discovery) or control siRNA (ON-TARGETplus Non-targeting Control Pool, D-001810-10-05, Horizon Discovery) using Opti-MEM medium (31985070, Gibco). Knock-down of *XPO1* mRNA and protein was verified after 24 h by RT-qPCR and immunoblot, respectively.

## Viral infections

**Infection of A549 cells to measure inhibition of viral titers.** A549 cells were grown on 6-well plates (3.5 cm plates) overnight. At a confluence around 90%, cells were pretreated with 100 μM 4OI, 1 μM SEL, 0.1 μM BARD, and 10 μM SFN for 12 h. They were then infected for 1 h with IAV PR8M at MOI = 0.05 in infection medium (DMEM medium containing pen/strep 1%, BSA 0.03%). After infection, cells were treated with the same concentrations of com-pounds in DMEM media containing pen/strep 1%, BSA 0.03% and TPCK-treated trypsin (1 μg/mL). Supernatants (for determination of viral titers) and cell pellets (for immunoblotting and RT-qPCR) were collected 12 and 24 h p.i.

**Immunofluorescence assay.** Immunofluorescence was performed to visualize the abun-dance and localization of NP in single cells as described before [44]. A549 cells were coated and grown on coverslips in 24 well plates overnight. At about 90% confluence, cells were pre-treated with 100 μM 4OI, 1 μM SEL, 0.1 μM BARD, and 10 μM SFN for 12 h and then infected with IAV (A/Puerto Rico/8/1934 (H1N1)) with MOI = 1 for 1 h in infection buffer (DMEM medium containing pen/strep 1%, BSA 0.03%). Cells were then incubated in fresh medium (DMEM medium with pen/strep 1%, BSA 0.03% and TPCK-treated trypsin [1 μg/mL]) con-taining the four compounds. At 4, 6 and 8 h p.i. the treated and untreated cells were washed three times with PBS and fixed with 500 μl 4% (v/v) paraformaldehyde at room temperature for 8 minutes. The cells were then washed three times with PBS and unspecific antibody

binding was blocked with 10% (w/v) BSA at room temperature for 1 h. Cells were then incubated with primary antibody (1:50 anti-NP mouse monoclonal IgG, hybridoma supernatant, kindly provided by Prof. Stephan Ludwig, Muenster, Germany) in PBS containing 1% BSA and 0.5% (v/v) Triton X-100) for 1 h at room temperature or for 16 h at 4˚C. After washing three times with PBS, the cells were incubated with goat anti-mouse IgG-Cy3 conjugated secondary antibody, which was diluted 1:3000 in PBS containing 1% BSA and 0.5% (v/v) Triton X-100, for 1 h at room temperature protected from light. Cells were then washed three times, and DNA was stained with DAPI (4',6-diamidino-2-phenylindole, Roth), diluted 1:400 in PBS. Coverslips were washed again and mounted on microscope slides with ProLong Gold Antifade Mountant (Thermo). Slides were stored at 4˚C while being protected from light. Further analysis was performed using a confocal laser scanning microscope (Leica TCS SP5, diode 405 nm for DAPI and DPSS 561 nm for NP).

For p53 detection, we followed the same procedure except that donkey serum containing 0.5% Triton X-100 was used for blocking and an Olympus FV3000 laser scanning microscope for imaging (405 nm for DAPI and 561 nm for p53). Unlabeled mouse anti-human p53 monoclonal antibody (X77, ThermoFisher Scientific cat no. MA1-12549) was used as primary antibody (dilution 1:300) and Alexa Fluor 568 labeled donkey anti-mouse IgG (H+L) (Invitrogen, cat no. A10037) as secondary antibody (dilution 1:1000).

**Mitochondrial ROS assay.** PR8M infection (MOI = 1) and treatments were carried out as described above. Upon conclusion of the experiment, the cells were incubated with medium containing 5 μM MitoSOX Red mitochondrial superoxide indicator (Invitrogen, cat# M36008) for 5 min. After washing with PBS, cells were resuspended with cold PBS for measurement of mitochondrial ROS by flow cytometry (Sony SP6800 ZE Analyzer, phycoerythrin channel).

**Infection of iPSC-derived EC.** Cells were grown in fibronectin coated plates in ECGM-2 medium as described above and infected with PR8M at MOI = 1. Treatments with the compounds and measurements of gene expression and viral titers were carried out as described above for A549 cells.

**Real-time quantitative reverse transcriptase polymerase chain reaction (RT-qPCR)** was performed as described in detail in [2], using Nucleospin RNA purification kit (Machery Nagel), on-column removal of DNA with rDNase (Machery Nagel), and the PrimeScript cDNA synthesis kit (TaKaRa, Shiga, Japan) with 400 ng RNA input in a 10 μl reaction. Sequences of PCR primers are shown in S2 Table. Relative expression of host target mRNA and viral HA mRNA was calculated using the $2^{-\Delta\Delta CT}$ and $2^{-\Delta CT}$ methods [45], respectively, using *HPRT* mRNA as internal reference.

**Immunoblotting** was performed as described in detail in [46], using a semi-dry transfer system (Trans-Blot Turbo, BioRad), Amersham enhanced chemiluminescence western blot detection reagent (GE Healthcare Science, Pittsburgh, USA), and a Vilber fusion FX7 device (Vilber Smart Imaging, Collégien, France). The following primary antibodies were used: influenza A virus nucleoprotein (NP, PA5-32242, 1:3000, ThermoFisher Scientific), XPO1 (46249S, 1:1000, Cell signaling technology), β-actin (ab49900, 1:20,000, Abcam). Goat anti-rabbit IgG-HRP (Southern Biotech, catalogue no. 4030–05) was used as secondary antibody.

**4OI-XPO1 pull-down assay.** The alkyne-tagged 4OI-probe (4-OI-alk) was synthesized according to the method described by Qi et al. [17] (S1 Methods). The human respiratory cell line Calu-3 was grown to 70% confluence in T75 (middle) flasks and treated with 400 μM 4OI-alk or DMSO for 4 h. The cells were washed twice with ice-cold PBS, harvested using cell scrapers and centrifuged at 400 g for 5 min at 4˚C. Cell pellets were then resuspended in 200 μL of 1% Triton X100 lysis buffer (1% Triton-X 100, 150 mM NaCl, 50 mM triethanolamine) and lysates centrifuged for 5 min (16 000 g, 4˚C) to remove debris. Protein

concentration was determined using a BCA protein assay kit. Then, lysates were equilibrated to 2 mg/mL and samples containing 200 μL lysate (400 μg protein) were prepared. To each sample was then added 6 μL 50 mM CuSO4 (Sigma-Aldrich, dissolved in MQ-H$_2$O), 14 μL 100 mM Tris(3-hydroxypropyltriazolylmethyl)amine (THPTA) ligand (TCI Chemicals, dissolved in MQ-H$_2$O), 6 μL 10 mM Biotin-Azide (Sigma-Aldrich, dissolved in DMSO) and 10 μL 100 mM Sodium Ascorbate (Sigma-Aldrich, dissolved in MQ-H$_2$O), followed by incubation for 1 h at room temperature. The resulting click-labeled lysates were precipitated with methanol overnight at -20˚C. Proteins were pelleted (7000 g, 5 min, 4˚C), washed through resuspension in 500 μL 9:1 MeOH:MQ-H$_2$O followed by re-pelleting, and finally resuspended in 1 mL PBS containing 0.5% SDS. At this point 20% of the protein solution was saved for the input control. The solutions were then incubated with 100 μl of streptavidin Dynabeads (Invitrogen) for 2 h at room temperature, followed by washing with PBS-T six times. The beads were then mixed with 40 μl loading buffer and heated to 95˚C for 5 min. Input and elution samples were resolved on 4–20% SDS-PAGE gel, and XPO1 and Keap1 proteins were detected via western blot using antibodies against XPO-1/CRM1 (D6V7N, rabbit mAb) andKEAP1 (D1G10) Rabbit mAb (both Cell Signaling Technologies). For competition experiments, cells were preincubated with unlabeled SEL (1, 4, 40 μM) for 30 min, and the 4OI click chemistry probe was then added to the medium for 2 h.

## Ligand-target modeling

All calculations were performed using the Molecular Operating Environment (MOE) version 2020.09 (https://www.chemcomp.com/Products.htm).

**Preparation of protein structures.** *XPO1/CRM1.* Three X-ray crystal structures of XPO1 were used in this study: complex with Selinexor (PDB ID: 7L5E) [20], complex with Leptomycin B (PDB ID: 4HAT) [18], complex with Leptomycin B (PDB ID: 6TVO) [21]. The potential was set up with Amber10:EHT as a force field and R-field for solvation. After removal of the co-crystalized ligand, addition of hydrogen atoms, removal of water molecules farther than 4.5 Å from ligand or receptor, correction of library errors, and tethered energy minimization of binding site were performed via the QuickPrep module. *KEAP1.* The X-ray crystal structure of the BTB domain of KEAP1 in complex with CDDO (PDB ID: 4CXT) [22] was used for the molecular-docking studies. The potential was set up with Amber10:EHT as a force field and R-field for solvation. After removal of the co-crystalized ligand, addition of hydrogen atoms, removal of water molecules farther than 4.5 Å from ligand or receptor, correction of library errors, and tethered energy minimization of binding site were performed via the QuickPrep module.

**Structural modeling.** The binding site was set to dummy atoms which were identified by the site finder command. Covalent docking was performed for the four compounds in the leptomycin B binding site of the three structures of XPO1 and the CDDO binding site of the BTB domain of KEAP1. Residues Cys528 and Cys151 were selected as the reactive site for XPO1 and KEAP1, respectively. Custom reaction files were created for each of the studied compounds as the reactions present in MOE by default were not suitable for our purposes. For BARD, two files were created in order to force the reaction center on either one of the two Michael acceptors (C1, C9) present in the molecule (Fig 1). For SEL, a custom file was created to deal with the *N*-substitution of the Michael acceptor moiety. For SFN, a custom file was created to force the cysteine sulfur to attack the electrophilic isothiocyanate carbon. In all cases, placement trials were set to 100 poses with an induced fit refinement. GBVI/WSA dG with 10 poses was used as final scoring function.

**Biostatistics.** All cellular experiments were set up in triplicates. Statistical analyses were performed using GraphPad Prism v8.0.2 (GraphPad Software), using one-way ANOVA with

correction for multiple testing to assess significance unless indicated otherwise in the figure legends. Data are expressed as means ± standard error of the mean (SEM) unless stated otherwise. The following abbreviations were used to indicate the level of statistical significance: *, $p < .05$; **, $p < .01$; ***, $p < .001$; ****, $p < .0001$.

## Supporting information

**S1 Fig. Percentage of infected cells throughout an 8 h time course of IAV infection.** A549 cells were pretreated with the compounds (SEL, 1 μM; 4OI, 100 μM; BARD, 0.1 μM; SFN, 10 μM) for 12 h, were then infected with IAV PR8M (MOI = 1) for 1 h and subsequently incubated in fresh medium containing the compounds. Analysis based on the same images as used for Fig 2C–2F. Total number of cells was determined by counting DAPI-positive nuclei, and IAV infected cells by counting cells staining positive for NP in nucleus, cytoplasm or both. Data correspond to averages from 7 microscopic fields. **A.** Percentage of infected cells at 4, 6, and 8 h p.i. One-way ANOVA with Tukey's post-hoc test. * ≤0.05, ** ≤0.01, *** ≤0.001, **** ≤0.0001.
(EPS)

**S2 Fig. Bar graphs corresponding to the heat map shown in Fig 3J.** The RT-qPCR data were analyzed by the $2^{-\Delta\Delta Ct}$ method using *HPRT* mRNA as internal control. Fold change was calculated with respect to expression in uninfected wild-type cells. $n = 3$, means ±SEM. One-way ANOVA with Tukey's post-hoc test. * ≤0.05, ** ≤0.01, *** ≤0.001, **** ≤0.0001.
(EPS)

**S3 Fig. Bar graphs corresponding to the heat map shown in Fig 4C.** The RT-qPCR data were analyzed by the $2^{-\Delta\Delta Ct}$ method using *HPRT* mRNA as internal control. Fold change was calculated with respect to expression in uninfected wild-type cells. $n = 3$, means ±SEM. One-way ANOVA with Tukey's post-hoc test. * ≤0.05, ** ≤0.01, *** ≤0.001, **** ≤0.0001.
(EPS)

**S4 Fig. The dynamic range of induction of anti-oxidative mRNAs by the four compounds is greater in iPSC-derived ECs than in A549 cells.** Reanalysis of the data of Figs 3J and 4C. **A-D,** Baseline expression of the target genes is significantly higher in A549 cells than in iPSC-derived ECs. RT-qPCR data were reanalyzed using the $2^{-\Delta Ct}$ method, using *HPRT* mRNA as reference. Differences in expression (expressed on $\log_2$ scale) in the absence of treatment was compared between A549 and iPSC-derived ECs, either in uninfected or infected cells. $n = 3$, means ±SEM. T-test. * ≤0.05, ** ≤0.01, *** ≤0.001, **** ≤0.0001. **E,** Induction of the target genes by the treatments is greater in iPSC-derived ECs than in A549 cells. Fold change (expressed as linear values) was computed for each cell type and compound by the $2^{-\Delta\Delta Ct}$ method, using infected untreated cells as reference. Differences between the two cell types in fold change due to the same treatment were assessed by T-test. * ≤0.05, ** ≤0.01, *** ≤0.001, **** ≤0.0001.
(EPS)

**S5 Fig. Effects of KEAP1 knock-down on *NRF2 (NFE2L2)* and *IFIT1* mRNA expression.** A549 cells were transfected for 24 h with specific siRNA targeting *KEAP1* mRNA or nonspecific siRNA. Cells were then pretreated with the compounds (SEL, 1 μM; 4OI, 100 μM; BARD, 0.1 μM; SFN, 10 μM) for 12 h, infected with IAV PR8M (MOI = 1) for 2 h, and then incubated in fresh buffer containing the compounds for 22 h. **A,B.** Efficiency of KEAP1 knock-down. **A.** *KEAP1* mRNA (RT-qPCR). **B.** KEAP1 protein (immunoblot). **C, D.** *NFE2L2* and *IFIT1* mRNA (RT-qPCR). n = 3, means ±SEM. One-way ANOVA with Tukey's post-hoc test, using

infected untreated wild-type or knock-down cells as reference. * $\leq$0.05, ** $\leq$0.01, *** $\leq$0.001, **** $\leq$0.0001.
(EPS)

**S6 Fig. Effects of *XPO1* knock-down in *NRF2*$^{-/-}$ ECs on antiviral activity of the compounds.** WT and *NRF2*$^{-/-}$ ECs were transfected for 24 h with specific siRNA targeting *XPO1* mRNA or nonspecific siRNA. Cells were then pretreated with the compounds (SEL, 1 μM; 4OI, 100 μM; BARD, 0.1 μM; SFN, 10 μM) for 12 h, infected with IAV PR8M (MOI = 1) for 2 h, and then incubated in fresh buffer containing the compounds for 22 h. **A.** *NRF2 (NFE2L2)* mRNA (RT-qPCR). **B.** *XPO1* mRNA (RT-qPCR), demonstrating a 75% knock-down. **C.** IAV titers in cell culture supernatants (foci-forming assay, FFU/mL). **D.** *IFIT1* mRNA (RT-qPCR). n = 3, means ±SEM. One-way ANOVA with Tukey's post-hoc test, using infected untreated wild-type or knock-down cells as reference. * $\leq$0.05, ** $\leq$0.01, *** $\leq$0.001, **** $\leq$0.0001.
(EPS)

**S7 Fig. Competition experiment demonstrating binding of 4OI and SEL to the same sites on XPO1 and KEAP1. A,B.** "Click-chemistry" pull-down assay demonstrating covalent binding of an alkynated 4OI probe (4-OI-alk) to XPO1 (A) and KEAP1 (B) in Calu-3 cells. Cells were preincubated with 1 or 4 μM unmodified SEL for 30 min. as indicated. Two hours after addition of the probe, proteins complexed with the probe were detected by immunoblot for XPO1 (A) or KEAP1 (B). **C,D.** Densitometry (arbitrary units) of the immunoblots, normalized to the signal obtained from the band labeled "input". SEL competes with 4OI for complex formation with both targets, suggesting that the compounds recognize the same sites on both targets. However, competition is less efficient for complex formation with KEAP1, suggesting that 4OI has higher affinity for KEAP1 and that SEL has higher affinity for XPO1.
(EPS)

**S8 Fig. Experiment identical as S6 Fig, but featuring a higher concentration of SEL.** SEL was added at concentrations of 4 and 40 μM. Due to a technical error, the signals in **B** (4 μM SEL) were higher than expected, but densitometry revealed a similar reduction in complex formation as in the experiment shown in S6 Fig. Densitometry could not be performed on **A** due to loss of part of the membrane (see missing lower border of band "Input 4 μM").
(EPS)

**S9 Fig. 3D structural modeling of 4OI-XPO1 interactions based on the co-crystal structure of XPO1 (CRM1) with leptomycin B (PDB ID: 6TVO).** Both 4OI and leptomycin B are covalently bound to the reactive Cys528 (marked with an asterisk *) and interact extensively with the hydrophobic NES-binding groove. **A.** 4OI binds the site through hydrophobic interactions between the octyl chain and Ile521, Leu525, Met545, Val565 and Leu569 in the hydrophobic pockets Φ2 and Φ3 of the NES-binding site. The C1-carboxyl group further stabilizes binding through two hydrogen bonds with Lys537 and Lys568. These hydrophobic and electrostatic interactions optimally direct the methylene group of 4OI towards Cys528 and could be the driving force for the covalent Michael 1,4-addition. **B.** Overlay of 4OI (cyan) and leptomycin B (magenta) in the NES-binding groove showing about 70% occupancy by leptomycin B and 40% by 4OI. Lipophilicity protein surface at the NES-binding cleft: lipophilic (green), hydrophilic (violet), neutral (white), α-helices (gold). * = Cys528.
(EPS)

**S10 Fig. The compounds favor nuclear retention of p53 in IAV-infected A549 cells.** A549 cells were treated and infected as described for Fig 2. p53 was detected by indirect immunofluorescence 8 h p.i., using Alexa Fluor 568 labeled secondary antibody. **A.** Representative

immunofluorescence images p53 = red. Nuclei = blue (DAPI). Pink signal in merged images = nuclear localized p53. The positive staining granular pattern is a technical artefact and was considered background signal. Negative control = no primary antibody. **B.** Fraction of all cells with nuclear p53 staining. Cells with nuclear p53 staining were counted by visual inspection by two independent examiners who were blinded to the identity of the specimens. n = 4 microscopic fields, means ±SEM. One-way ANOVA with Tukey's post-hoc test, using infected untreated wild-type or knock-down cells as reference. * ≤0.05, ** ≤0.01, *** ≤0.001, **** ≤0.0001.
(EPS)

**S1 Table. Viruses with pharmacologic evidence of XPO1 (CRM1)-dependence.**
(XLSX)

**S2 Table. List of RT-qPCR primers.**
(XLSX)

**S1 Methods. NRF2 activators inhibit influenza A virus replication by interfering with nucleo-cytoplasmic export of viral RNPs in an NRF2-independent manner.**
(DOCX)

## Author Contributions

**Conceptualization:** Fakhar H. Waqas, Mahmoud Shehata, Walid A. M. Elgaher, Thomas B. Poulsen, Ulrich Martin, Ruth Olmer, David Olagnier, Anna K. H. Hirsch, Stephan Pleschka, Frank Pessler.

**Data curation:** Fakhar H. Waqas, Mahmoud Shehata, Walid A. M. Elgaher, Antoine Lacour.

**Formal analysis:** Fakhar H. Waqas, Mahmoud Shehata, Walid A. M. Elgaher, Antoine Lacour, Naziia Kurmasheva, Andreas Pavlou, Thomas B. Poulsen.

**Funding acquisition:** Mahmoud Shehata, Ulrich Martin, Ruth Olmer, Stephan Pleschka, Frank Pessler.

**Investigation:** Fakhar H. Waqas, Mahmoud Shehata, Walid A. M. Elgaher, Antoine Lacour, Naziia Kurmasheva, Fabio Begnini, Anders E. Kiib, Julia Dahlmann, Chutao Chen, Thomas B. Poulsen, Sylvia Merkert, Ruth Olmer, David Olagnier.

**Methodology:** Fakhar H. Waqas, Mahmoud Shehata, Walid A. M. Elgaher, Antoine Lacour, Naziia Kurmasheva, Fabio Begnini, Anders E. Kiib, Julia Dahlmann, Thomas B. Poulsen, Sylvia Merkert, Ruth Olmer, David Olagnier, Stephan Pleschka.

**Project administration:** Fakhar H. Waqas, Chutao Chen, Anna K. H. Hirsch, Stephan Pleschka, Frank Pessler.

**Resources:** Andreas Pavlou, Thomas B. Poulsen, Sylvia Merkert, Ulrich Martin, Ruth Olmer, Anna K. H. Hirsch, Stephan Pleschka, Frank Pessler.

**Supervision:** Walid A. M. Elgaher, Andreas Pavlou, Thomas B. Poulsen, Ulrich Martin, Ruth Olmer, David Olagnier, Anna K. H. Hirsch, Stephan Pleschka, Frank Pessler.

**Visualization:** Fakhar H. Waqas, Mahmoud Shehata, Walid A. M. Elgaher, Antoine Lacour, Andreas Pavlou.

**Writing – original draft:** Fakhar H. Waqas, Mahmoud Shehata, Walid A. M. Elgaher, Antoine Lacour, Naziia Kurmasheva, Thomas B. Poulsen, David Olagnier, Frank Pessler.

**Writing – review & editing:** Fakhar H. Waqas, Mahmoud Shehata, Walid A. M. Elgaher, Antoine Lacour, Naziia Kurmasheva, Thomas B. Poulsen, Ruth Olmer, David Olagnier, Anna K. H. Hirsch, Stephan Pleschka, Frank Pessler.

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
