## [Decision Letter · Decision Letter 0]

14 Mar 2023

Dear Dr. Pessler,

Thank you very much for submitting your manuscript "NRF2 activators inhibit influenza A virus replication by interfering with nucleo-cytoplasmic export of viral RNPs in an NRF2-independent manner" for consideration at PLOS Pathogens. As with all papers reviewed by the journal, your manuscript was reviewed by members of the editorial board and by several independent reviewers. In light of the reviews (below this email), we would like to invite the resubmission of a significantly-revised version that takes into account the reviewers' comments.

The reviewers offer several ways in which to strengthen and improve the manuscript conclusions. Please address all of the reviewer comments in a revised manuscript. Experiments with an influenza virus minigenome system are encouraged but would not be required for resubmission.

We cannot make any decision about publication until we have seen the revised manuscript and your response to the reviewers' comments. Your revised manuscript is also likely to be sent to reviewers for further evaluation.

Sincerely,

Jacob S. Yount

Academic Editor

PLOS Pathogens

Meike Dittmann

Section Editor

PLOS Pathogens

Kasturi Haldar

Editor-in-Chief

PLOS Pathogens

orcid.org/0000-0001-5065-158X

Michael Malim

Editor-in-Chief

PLOS Pathogens

orcid.org/0000-0002-7699-2064

The reviewers offer several ways in which to strengthen and improve the manuscript conclusions. Please address all of the reviewer comments in a revised manuscript. Experiments with an influenza virus minigenome system are encouraged but would not be required for resubmission.

Reviewer's Responses to Questions

**Part I - Summary**

Reviewer #1: The life cycle of influenza A virus (IAV) involves intricate interplay with host cellular factors. Due to the drug-resistance issue of current available anti-IAV drugs that all target viral proteins, it will be highly desired to develop host-directed antiviral drugs. In addition to the well-characterized antioxidative and anti-inflammatory effects of NRF2 activators, they also broadly interfere with the replication of RNA and DNA virus, including IAV. However, the exact mechanism of the antiviral activity of NRF2 activators is largely unclear. The authors in this study endeavored to clarify the underlying anti-IAV mechanisms of four compounds (4OI, BARD, SFN, SEL). They found that although acting as NRF2 activators, the antiviral effect of these compounds on IAV replication was NRF2-independent. Instead, their anti-IAV activity was correlated with their ability to inhibit the nuclear export of vRNP complex. Structural modeling predicted covalent binding of all three NRF2 activators and SEL to the active site of XPO1 critical amino acid Cys528. They concluded that the NRF2 activators inhibit IAV replication by interfering with the nuclear export of viral RNPs in an XPO1-dependent manner. However, some comments have to be addressed to improve the manuscript.

Reviewer #2: In this manuscript, Waqas et al. seek to shed light on the mechanisms regulating the antiviral effects of three NRF2 activators on IAV. They show that 4OI, SFN and BARD, as well as the potent CRM1 inhibitor SEL, inhibit more or less efficiently the nuclear export of IAV vRNPs. Surprisingly, they show that the compounds target, interact and probably inactivate CRM1, and that this inhibition is NFR2 independent. More interestingly, they provide ligand-target modeling data suggesting that the three NRF2 activators interact with different energies with the XPO1 NES-binding hydrophobic pocket. Moreover, the XPO1 inhibitor SEL can potentially interact in the same way with the active site of KEAP1, which is the main target of 4OI, SFN and BARD.

**Part II – Major Issues: Key Experiments Required for Acceptance**

Reviewer #1: 1. The authors examined three NRF2 activators, and found that the anti-IAV activity was correlated with their binding energies to XPO1. Are there any other NRF2 activators, if so, whether their anti-IAV activity all depends on their ability to bind XPO1 rather than acting as NRF2 activators. Is the antiviral effect of these NRF2 activators specific to IAV? How about their effect on other viruses that do not have a phase of replication in the nucleus?

2. The authors demonstrated the inhibitory effect of these NRF2 activators on the nuclear export of vRNP complex of IAV. How about their effect on the nuclear export of host cellular proteins. To exclude potential toxic effect, the safe concentration range of the compounds in cells should be provided.

3. To further confirm the conclusion that the four compounds inhibit nuclear export of vRNP complex, the authors had better exclude the possibility that these compounds may affect the vRNP complex activity of IAV by using a minigenome assay.

Reviewer #2: 1) Experiments and Figure 3J:

- As NRF2 activators likely display their effects by binding and inactivating KEAP1, the expression of the mRNA of the latter should also be checked and provided in the heat map for the different conditions.

2) Experiments and Figure 4:

- The authors show that the NRF2 KO cells support significantly higher viral titers, which are still more or less reduced in the presence of the four compounds. They should again analyse the mRNA amounts of KEAP1 and if they remain unchanged, the authors should knock-down KEAP1 with specific siRNAs and then evaluate the virus titers. According to the results provided in Figure 5 and Figure 6, the authors claim that unlike SEL, BARD is a better ligand for KEAP1 than for XPO1 and if so, in the absence of KEAP1, BARD might become more available for binding and inhibiting XPO1 and would thus reduce more efficiently the viral titers.

- Analysing the viral titers in conditions of NRF KO and XPO1 KD will provide additional data on the mechanism of action of the analysed compounds.

3) Experiments and Figure 5A & B:

- If 4OI binds, as does SEL, the same active sites on XPO1 and KEAP1, competition experiments between these two compounds should provide additional proofs.

**Part III – Minor Issues: Editorial and Data Presentation Modifications**

Reviewer #1: 1. The shuttling of vRNP complex between cytoplasm and nucleus is critical for the replication of IAV. To offer informative background information, the authors could present the current research advances in the nuclear import and export of vRNP complex of IAV, especially the host factors involved except for XPO1.

2. Page 5, Lines 143-145: Knock-down of XPO1 did not affect levels of IAV hemagglutinin (HA) mRNA or NP in the absence or presence of the four compounds (Fig. 3D, E). However, in Figure 3E, the intensity of ACTB in lanes 8-10 is lower than those in other lanes. Meanwhile, the expression of NP in SEL-treated XPO1 KD cells is also obviously lower than others. Are the provided images representative? The ratio of NP/ACTB had better to be presented by using the ImageJ software.

3. All compounds reduced intracellular IFN responses. The lower mRNA level of IFIT1 and CXCL10 in the drug treatment group can be associated with the inhibition of IFN pathway by drug itself or the lower virus titer caused by the compound treatment. The ISRE reporter assay had better to be carried out to differentiate these possibilities.

4. Line 360: the authors carried out the immunofluorescence assay at 2, 4, 6 and 8 h p.i. However, the images of NP staining at 2 h p.i. were not shown in Fig. 2C.

5. How was the virus titer determined in Fig. 3F and 4B? What does FFU mean?

Reviewer #2: 1) Experiments and Figure 2:

- The authors should explain the huge differences in the concentrations of the four compounds used, varying from 0.1 to 100 μM and which, I believe, are related to their cellular toxicity.

- In 2C the authors should indicate and discuss the percentage of infected (NP-expressing) cells for each condition, with or without treatment.

- IF staining of the different conditions is also needed at 0 hpi, as control.

Minor issues:

The authors should carefully check the references: references 1 and 14 are identical, the reference Qin et al. in the Supplemental Data is already cited in the main list as n°18…

PLOS authors have the option to publish the peer review history of their article (what does this mean?). If published, this will include your full peer review and any attached files.

Reviewer #1: No

Reviewer #2: No
---

## [Editor Report · Decision Letter 1]

22 Jun 2023

Dear Dr. Pessler,

We are pleased to inform you that your manuscript 'NRF2 activators inhibit influenza A virus replication by interfering with nucleo-cytoplasmic export of viral RNPs in an NRF2-independent manner' has been provisionally accepted for publication in PLOS Pathogens.

Best regards,

Jacob S. Yount

Academic Editor

PLOS Pathogens

Meike Dittmann

Section Editor

PLOS Pathogens

Kasturi Haldar

Editor-in-Chief

PLOS Pathogens

orcid.org/0000-0001-5065-158X

Michael Malim

Editor-in-Chief

PLOS Pathogens

orcid.org/0000-0002-7699-2064
---

## [Editor Report · Acceptance letter]

12 Jul 2023

Dear Dr. Pessler,

We are delighted to inform you that your manuscript, "NRF2 activators inhibit influenza A virus replication by interfering with nucleo-cytoplasmic export of viral RNPs in an NRF2-independent manner," has been formally accepted for publication in PLOS Pathogens.

Best regards,

Kasturi Haldar

Editor-in-Chief

PLOS Pathogens

orcid.org/0000-0001-5065-158X

Michael Malim

Editor-in-Chief

PLOS Pathogens

orcid.org/0000-0002-7699-2064